# Evaluation of Processing Conditions and Hydrocolloid Addition on Functional Properties of Aquafaba

**DOI:** 10.3390/foods12040775

**Published:** 2023-02-10

**Authors:** Kaelyn Crawford, Catrin Tyl, William Kerr

**Affiliations:** 1Department of Food Science and Technology, University of Georgia, Athens, GA 30602, USA; 2Faculty of Chemistry, Biotechnology and Food Science, Norwegian University of Life Science, 1432 Ås, Norway

**Keywords:** aquafaba, foams, food processing, physical properties

## Abstract

Aquafaba, the cooking water from chickpeas, could replace animal-derived ingredients such as egg whites in systems that require the stabilization of an oil or gas phase. However, little is known about how processing methods and additives affect its functional properties. In this study, aquafaba was prepared via boiling or pressure-cooking at water-to-seed (WSR) ratios of 5:1, 4:1 and 3:1. The effects of preparation method and pH adjustment on viscosity, protein content, solubility and profile were evaluated. Samples were further analyzed for foaming capacity/stability (FC/FS) and emulsifying activity/stability index (EAI/ESI). Foams were also prepared in combination with xanthan gum or hydroxypropyl methylcellulose (HPMC). Solubility was lowest near pH 4 and not affected by cooking method and protein profile was not affected by method or ratio. Samples with pH 3 had high EAI and FS, but low ESI and FC. WSR did not significantly affect interfacial properties. Xanthan gum had a greater effect than HPMC on viscosity and prevented foam liquid drainage for 24 h. While the preparation method affects aquafaba properties, subsequent pH adjustment is of greater relevance for interfacial properties. Foam volumes can be maximized and foam drainage limited by appropriate choice of hydrocolloids and addition levels.

## 1. Introduction

Food by-products and waste are a concern of producers and consumers alike and discovering innovative uses for these waste streams within the food industry is both a desired and necessary development. With the current demand for plant protein only expected to increase, it would be beneficial to exploit existing waste streams for this purpose. Canning is an integral unit operation in the preparation of certain pulse products and generates large quantities of cooking water, the so-called aquafaba. Interest in this material is based on its interfacial properties, which allow for the stabilization of dispersed systems. Initial explorations of its use were led by culinary professionals as well as the vegetarian and vegan community. Recently, scientific studies have reported on aquafaba’s foaming and emulsifying properties under multiple conditions and preparatory methods. Buhl et al. [1] found that aquafaba could act as a foaming agent. While it had lower foaming capacity than egg white powder at pH 6, it did provide better emulsion stability. Nguyen et al. [2] studied the effects of various conditions on the texture and microstructure of foams and found that optimum foam volume and stability were attained with 4:1 (water:bean) cooking ratio and addition of 0.7 g·mL^−1^ sucrose, 0.003 g·mL^−1^ NaCl and 0.00005 g·mL^−1^ xanthan gum. Subsequent work by this group Nguyen et al. [3] showed that aquafaba could be used as an egg replacer in cake when adjusted to pH 4 and 3 mg·mL^−1^ NaCl was included. The inclusion of xanthan gum helped improve the foam stability, hardness and bubble size in the foam. Some work has also been done on aquafaba not made from chickpea. Mousalvi [4] found that barley aquafaba could be used to partially replace egg white in oily cake. That is, replacement of up to 50% of egg white led to cake properties similar to control. However, higher levels of the barley aquafa decreased the cake volume. Arozarena et al. [5] researched the replacement of egg white in yellow cake by lupine aquafaba. While many of the cakes formed and collapsed, an acceptable cake could be made by adding baking powder, mono- and di-glyceride emulsifiers, 0.55% xanthan gum, plus sugar.

While several studies have optimized cooking conditions, there is still only a limited knowledge regarding the effect of processing conditions on functional properties. Moreover, the food industry often uses several ingredients to ensure and prolong the stability of dispersions. In particular, hydrocolloids are extensively used to reduce detrimental processes such as coalescence or creaming. However, research on adding hydrocolloids to aquafaba has so far been limited, with Nguyen et al. [2] evaluating the effect of xanthan gum (XG) on chickpea aquafaba foams with low sugar levels as well as the additional factor of sea salt. These authors used foaming capacity (FC), foaming stability (FS) and foam hardness to select the optimum level of each factor (cooking ratio, pH, sugar quantity, salt quantity, XG quantity). However, the cooking ratio was not factored into the influence on foam texture at multiple pH values and sugar levels.

The purpose of this research was a comprehensive evaluation of aquafaba’s interfacial and rheological properties in relation to processing conditions and presence of additional ingredients. We simultaneously assessed the effects of cooking method, water:seed ratio, pH and hydrocolloid gum addition on emulsion and foaming properties afforded by aquafaba to expand and connect factors explored in previous research. Initial experiments focused on the effects of processing (boiling versus pressure-cooking), water:seed ratio (3:1 to 5:1) and pH (3–8) on the types of proteins extracted, the protein concentration, the viscosity and the solubility of proteins in the aquafaba. From there, some interfacial properties of the aquafaba were assessed. To determine how the aquafaba would function in a food system, foams were formed using optimized aquafaba samples which were combined with sugar and various levels (0–0.6%) of either of two hydrocolloids (XG and hydroxypropyl methyl cellulose, HPMC). Rheological properties of these blends were analyzed, followed by the preparation of foams and measurement of foam overrun, the rate of liquid drainage from the foam lamella and foam firmness.

## 2. Materials and Methods

### 2.1. Sample Preparation

Kabuli Sierra garbanzo beans (36 kg) were purchased from Palouse Brand chickpeas (Palouse, WA, USA). All chickpeas were stored frozen (−14 °C) and used within one year. Samples were prepared using the method of Stantiall et al. [6]. Before cooking, chickpeas were soaked in tap water for 16 h at 2 °C in a sealed container and at a ratio of 3.3:1 g H_2_O/g seed. The soaking water was then drained and discarded. Soaked chickpeas were rinsed with tap water, patted dry and stored in a sealed container at 2 °C until cooked.

### 2.2. Sample Processing

After soaking, the chickpeas were cooked by either boiling or pressure-cooking (Figure 1). Tap water and chickpea seed were weighed out to create cooking ratios of 3:1, 4:1, or 5:1 g H_2_O/g seed. For samples prepared by boiling, water was added to a cooking pot and heated at 100 °C until boiling began. The chickpea seeds were then added into the boiling water and cooked for 1 h. For samples prepared by pressure cooking, tap water and chickpea seed was added to the 2 L pressure-cooking vessel (Hawkins, Mumbai, Maharashtra, India) and placed on a hot plate (Rosewill, Shanghai, China) at its maximum heat setting (121 °C) until steam exited the open valve. To reach full pressure in the vessel a cap was then placed to seal the valve and the maximum heat setting was maintained until the cap lifted to release steam. The temperature was then lowered to 93 °C and the chickpeas were cooked for 5 min. After 5 min, the pressure-cooking vessel was removed from the heat source and allowed to depressurize for 10 min. Once cooking was completed, chickpea seeds and the cooking liquid were transferred to sealed mason jars and steeped for 24 h at room temperature (20 °C). After 24 h, the chickpea seeds were strained and the cooking liquid (aquafaba) was stored in plastic bottles at −14 °C.

### 2.3. Adjusting Aquafaba pH

Based on existing aquafaba studies and preliminary work, pH values of pH 3, 5 and 7 were selected to analyze foaming and emulsifying properties. In addition, these were compared to the foaming and emulsifying properties of each sample at its native pH (6). Aquafaba samples prepared by both cooking methods were defrosted overnight at 4 °C and up to 2 h at room temperature if necessary. The method for changing sample pH was based on that of Cui et al. [7]. Samples were stirred using magnetic stir bars at room temperature while the pH was changed to one of the three values selected over a 1 h period, checking every 15 min with adjustment as necessary to keep the pH constant. The samples were then centrifuged at 11,000× *g* for 10 min and the supernatant separated for further analyses.

### 2.4. Protein Concentration and Protein Profile

The protein concentration of samples was determined using the Bradford assay [8]. The protein profile of each sample was determined via SDS-PAGE using a modified method by Alsalman & Ramaswamy [9]. Samples were defrosted at 2 °C overnight and at room temperature for up to 2 h if necessary. All samples were then standardized to 0.5 g/mL by diluting with phosphate buffer. Samples were prepared in both reducing and non-reducing conditions. In non-reducing conditions, 10 µL of standardized sample (0.5 g/mL) and 20 µL Laemmli buffer were added to microcentrifuge tubes. In reducing conditions, 10 µL of standardized sample (0.5 g/mL) and 19 µL Laemmli buffer were added along with 1 µL β-mercapto-ethanol. The tubes were boiled for 5 min, cooled and centrifuged at 14,000 RPM in a Sorvall RC 6 Plus Centrifuge (Thermo Fisher Scientific, Oslo, Norway) for 10 min. Next, 10 µL of each supernatant was loaded onto 4–20% polyacrylamide Tris-Glycine precast gels (Bio-Rad, Berkley, CA, USA), along with 7 µL of Precision Plus Protein Standard (Bio-Rad, Berkley, CA, USA) with molecular weights from 10–250 kDa. The proteins were separated using 10× Tris/Glycine/SDS running buffer on a Mini-PROTEAN Tetra cell (BIO-RAD, Hercules, CA, USA) at 100 V for 1.5 h. The gels were stained with Coomassie brilliant blue reagent for 45 min before de-staining with ultrapure water overnight. After de-staining, gels were scanned using an Epson flat-bed scanner (Los Alamitos, CA, USA). Bands were assigned using ImageJ 1.50i version software (National Institute of Health, Rockville, MD, USA) to estimate molecular weights in comparison to the protein standard bands.

### 2.5. Viscosity

Instrument rheological analysis of all aquafaba samples was performed using the methods of Alsalman et al. [10] with modification. Aquafaba samples were evaluated post-steeping and before freezing. A Discovery HR-2 hybrid rheometer (TA Instruments, New Castle, DE, USA) with TRIOS 5.2 software (TA Instruments, New Castle, DE, USA) was used for analyses. For each replicate, approximately 2 mL of samples were dispensed onto the rheometer plate. A 60 mm 2.0° cone (TA Instruments, Elstree, UK) was lowered to the trim gap at 105 µm, where the sample amount was adjusted if necessary to extend to the edge of the cone. The cone was then lowered to the final gap of 52 µm above the plate. Samples were then evaluated using a linear flow rate test. First, samples went through a conditioning step of 20 s soak time and a 60 s period of pre-shear. Next, samples underwent a shear flow ramp of 0.1 s^−1^ to 100 s^−1^ over an 840 s period, while the temperature was maintained at 25 °C and points sampled every 5 s. Each treatment group was analyzed in triplicate, with three replicates of each sample. A power law model was fit to the shear stress versus shear rate results using TRIOS 5.2 software.

### 2.6. Solubility

The protein solubility was analyzed by taking aliquots of boiling or pressure-cooked aquafaba samples from one cooking ratio (1:5 g seed/g H_2_O). Samples of 30 and 50 mL respectively were adjusted to pH 2 with 0.1 M HCl dispensed by a HI 901 potentiometric auto-titrator (Hannah Instruments, Smithfield, RI, USA). Aliquots of the same quantity were adjusted to pH 3, 4, 5, 6, 7 and 8 ± 0.1 using 0.1 M HCl or 0.1 M NaOH and standardized to the volume of the pH 2 sample by adding DI water. Samples were then centrifuged at 11,000× *g* for 10 min, with the resulting supernatant being analyzed using the Bradford assay [8].

### 2.7. Foaming Capacity and Stability

Foaming capacity and stability of aquafaba samples were performed using the method of Aslan & Ertas [11] with modifications. Thirty mL of liquid (*V*_i_) in a 250 mL beaker was foamed for 2 min at 10,000 RPM using a Polytron 2500 E homogenizer (Kinemtica Inc., Bohemia, NY, USA). Foams were transferred to 250 mL graduated cylinders and their volumes were measured at 0 min (*V*_0_) and 30 min (*V*_30_). Foaming capacity and stability were calculated using Equations (1) and (2), respectively:(1)$$\mathrm{Foaming}\;\mathrm{capacity}\;(\%)=\frac{V_{0}}{V_{i}}\times 100$$
(2)$$\mathrm{Foaming}\;\mathrm{stability}\;(\%)=\frac{V_{30}}{V_{i}}\times 100$$

### 2.8. Emulsifying Activity Index and Emulsifying Stability Index

Emulsifying activity and stability index of samples were analyzed following the original method of Pearce and Kinsella [12] as modified by [1]. Emulsions were prepared by using a high-speed homogenizer (Kinematica Inc., Bohemia, NY, USA) to blend 10 mL of vegetable oil into 30 mL of aquafaba at 8000 RPM for 5 min followed by 13,500 RPM for 4 min. An aliquot (200 µL) of the freshly prepared emulsion was pipetted into 50 mL of 0.1% w/v sodium dodecyl sulphate (SDS). Immediately after dilution, the absorbance was measured at 500 nm using a Genesys 150 UV-Visible spectrophotometer (Thermo Fisher Scientific, Waltham, MA, USA). Emulsifying activity index (EAI) was calculated using Equation (3). The equation to calculate EAI was from [1,13] with turbidity (*T*) and oil volume fraction (ϕ) calculated using equations outlined by He et al. [14]: (3)$$\mathrm{Emulsifying}\;\mathrm{activity}\;\mathrm{index}\;(\mathrm{EAI})=\frac{2T}{c\times \left(1-\phi \right)\times 10,000}$$
where *T* = turbidity in 1/m, *ϕ* is the oil volume fraction of the dispersed phase and *c* is the concentration of aquafaba protein in the emulsion. The turbidity (*T*) was calculated using Equation (4):(4)$$T=\frac{2.303\times A_{0}\times V}{I}$$
where *V* = dilution factor, *A*_0_ = absorbance of 500 nm at 0 min and *I* = path length (0.01 m). The oil volume fraction was calculated using Equation (5): (5)$$\phi =\frac{C-A-E\left(B-C\right)}{C-A+\left\{\frac{\left(B-C\right)\left(1+E\right)D_{0}}{D_{s}}-E\right\}}$$
where *A* is the beaker mass, *B* is the beaker mass plus 1 mL emulsion, *C* is the beaker mass plus the emulsion dry matter, *D*_0_ is the oil density, *D_s_* is the aquafaba solution density and *E* is the concentration of protein (mg protein/mg solvent (water)).

The emulsifying stability index (ESI) was determined by measuring absorbance after 10 min and calculated using Equation (6):(6)$$\mathrm{Emulsifying}\;\mathrm{stability}\;\mathrm{index}=\frac{A_{0}}{A_{0}-A_{10}}\times t$$
where *A*_0_ = absorbance of 500 nm at 0 min, *A*_10_ = absorbance of 500 nm at 10 min and *t* = time (10 min).

### 2.9. Foam Preparation

In addition to measurements on the base liquid, foams were prepared to compare overrun, liquid drainage and foam texture of boiled and pressure cooked aquafaba at 3:1 water to seed ratio with a pH of 5 and with and without addition of xanthan gum (XG) or hydroxypropyl methylcellulose (HPMC) at 3 levels (0.2%, 0.4% and 0.6%). Samples were adjusted to pH 5 over the course of 1 h while stirring at room temperature (18 °C) and checking every 15 min to ensure pH was constant. Samples were centrifuged at 11,000× *g* for 10 min in a low-speed Eppendorf centrifuge (Eppendorf, Enfield, CT, USA) and the supernatant separated. Next, 60 mL of the supernatant was added to a KitchenAid stand mixer (KitchenAid, St. Joseph, MI, USA) with a 5-quart stationary bowl and a stainless-steel chefs whisk beater. If testing XG or HPMC levels, gum was added prior to whipping and mixed in at speed 1. Foams were then whipped at speed 10 for 7 min, then 86.4 g of sugar were slowly added while whipping for 1 min at speed 3. 

### 2.10. Foam Overrun and Foam Liquid Drainage

Foam overrun (FO) was measured as described by Mohanan et al. [15] with some modification. Immediately after whipping, foams were transferred from the stand mixer bowl into 4 tared 50 mL weigh boats. The weigh boats were levelled using a metal dough divider for consistent measurement. The foam weight was recorded and the average of the four measurements was used to calculate foam overrun using Equation (7):(7)$$\mathrm{Foam}\;\mathrm{overrun}\;(\%)=\frac{W_{50\;\mathrm{mL}\;\mathrm{liquid}}-W_{50\;\mathrm{mL}\;\mathrm{foam}}}{W_{50\;\mathrm{mL}\;\mathrm{foam}}}*100$$
where *W*_50 mL liquid_ is the weight of an equal volume of aquafaba liquid used to make foams. Foam liquid drainage was measured according to Meurer et al. [1,16] with some modification. The total foam formed from 60 mL aquafaba was transferred from the stand mixer bowl immediately after whipping to 150 mL plastic funnels placed on 150 mL graduated cylinders. Gauze had been placed in the funnel to help contain the foam. The drip volume in mL was recorded at 0, 1 and 24 h.

### 2.11. Foam Texture

Hardness and adhesiveness of aquafaba foams were analyzed using the procedure described by [2,16]. After whipping, a cylindrical mold (diameter 50 mm, height 100 mm) was filled with foam and leveled with a metal dough divider. A TA-XT2 texture analyzer (Stable Micro Systems, Godalming, Surrey, UK) was used to assess foam texture using a compression test performed with a 35 mm diameter cylindrical probe with 50% deformation. Hardness was measured as the peak force value and adhesiveness as the negative area under the curve. Samples were tested in triplicate. 

### 2.12. Egg White and Gum Solution Viscosity

The viscosity of aquafaba solutions at pH 5 with and without addition of XG and HPMC (0 to 0.6% by mass) and 86.4 g sugar was analyzed using the methods of [10] with modification. In addition, the viscosity of egg white (EW) with 86.4 g sugar was analyzed for comparison. All samples were stirred using magnetic stir bars at room temperature for 1 h to allow solubilization of sugar and gums. Prior to analysis, samples were sonicated using an ultrasonic bath (Cole-Parmer, Vernon Hills, IL, USA) for 20 min to remove bubbles formed during mixing. A Discovery HR-2 hybrid rheometer (TA Instruments, New Castle, DE, USA) with TRIOS software 5.2 (TA Instruments, New Castle, DE, USA) was used for analyses. Approximately 2 mL of samples were dispensed onto the rheometer plate, then a 60 mm 2.0° cone (TA Instruments, UK) was lowered to the trim gap at 105 µm where sample amount was adjusted if necessary to extend to edge of the probe. The cone was lowered to the geometry gap at 52 µm and samples were then tested using a linear flow rate test at 25 °C. The samples went through a conditioning step with a 20 s soak time and a 60 s period of pre-shear. Next, samples underwent a flow ramp at 25 °C while shearing occurred over 840 s at a rate of 0.1 s^−1^ to 100 s^−1^, with measurements recorded every 5 s. Samples were analyzed in triplicate. A power law model was fit with the TRIOS software.

### 2.13. Statistical Analysis

All samples were tested in triplicate (n = 3) with aquafaba sample viscosities also measured three times per sample in triplicate (n = 9). Analysis was conducted in R (R Foundation for Statistical Computing, Vienna, Austria). Two-way analysis of variance (ANOVA) was used for analyses with two factors (cooking method and cooking ratio) while three-way ANOVA was used for analyses with three factors (cooking method, cooking ratio and pH or cooking method, gum type and gum level). Tukey’s honest significant difference (HSD) test was also performed for significant difference at *p* < 0.05. 

## 3. Results

### 3.1. Viscosity, Protein Content and Profile of Aquafaba

Table 1 shows the protein concentration of egg white and aquafaba samples for boiled and pressure cooked aquafaba samples at ratios of 5:1, 4:1 and 3:1 (water to seed). The protein concentration of boiled aquafaba at a ratio of 4:1 (1.4 mg/mL) was the highest of all samples. Overall, boiled samples had significantly greater (*p* ≤ 0.05) protein concentrations than pressure cooked samples, with B4:1 (boiled at 4:1 water:seed ratio) average protein concentration being significantly greater (*p* ≤ 0.05) than P4:1 and P5:1. This is likely due to pressure-cooked samples having only lost water when the pressure was released, while boiled samples continuously lost water via steam through a hole in the lid. The seed-to-water ratio was not a significant factor for protein concentration. We note that this represents soluble protein so values may be somewhat lower than others reported in the literature (such as [17]) that show total protein in the aquafaba.

The apparent viscosity (mPa·s) is calculated as the shear rate approaches zero and ranged from 9.6 to 81 mPa·s (Table 1). The boiled samples with a 3:1 ratio had significantly (*p* ≤ 0.05) higher apparent viscosity than all other samples, followed by the boiled sample at a 4:1 ratio. None of the other samples differed from each other in terms of apparent viscosity. Viscosity has been shown to have a strong correlation to total soluble solids and protein content in aquafaba [6,18]. Both cooking method and cooking ratio were significant factors. This could be related to total soluble solids, although this was difficult to prove as the protein concentration of different cooking ratios was not significantly different.

The flow behavior index (n) describes the shear-thinning behavior, with n = 1 indicating a Newtonian fluid and n < 1 indicative of pseudoplastic fluids [19]. The flow behavior index ranged from 0.66 to 0.95, with pressure cooked samples at a 5:1 ratio having the highest flow behavior index and samples boiled at a 3:1 ratio having the lowest. Samples with lower ratios had lower n values thus more shear-thinning behavior, as well as greater viscosity. This is consistent with the results of [10] that demonstrated decreased flow behavior index and likewise increased pseudoplasticity, when the cooking ratio of pressure cooked aquafaba increased.

SDS-PAGE gels of boiled and pressure cooked aquafaba samples are shown in Figure 2 for samples analyzed under non-reducing and reducing conditions, respectively. The bands ranged from 10 to 85 kDa with a similar distribution and intensity of bands across all samples, indicating that different cooking methods and cooking ratios did not change the protein fractions present. Based on comparison with literature data, these bands were tentatively assigned to 2S albumin (10nd 12 kDa), the γ-subunit of 7S vicilin (16 kDa), an unspecified 11S legumin-type protein (20 kDa), the basic and acidic subunit of 11S legumin (24 and 35 kDa, respectively) and a 7S vicilin precursor (50 kDa), with the band at 85 kDa uncharacterized. This corresponds to the protein profiles of chickpea aquafaba determined by [1,9,20,21,22]. These studies on chickpea aquafaba all recorded proteins with a molecular mass less than 48 kDa except for [21] who identified bands at 92.1 and 92.9 as O-acyltransferase and tRNA (Cystosine-5-)-methyltransferase. Notably, Alsalman et al. [9] was the only study to characterize the cooking water of self-prepared aquafaba, whereas other authors used canned aquafaba. Additionally, samples analyzed by [9] were pressure-cooked for 60 min, significantly longer than the cooking time of the pressure-cooked samples used in our study. The longer cooking time under harsh conditions could have caused changes in the protein pattern. Gels obtained under reducing conditions did not reveal additional bands to those present on gels that were run under non-reducing conditions, as also observed by [1]. As discussed by [1], this may be due to a lack of disulfide bonds among proteins in aquafaba.

### 3.2. Protein Solubility

Figure 3 shows the solubility curve of pressure cooked and boiled aquafaba samples from pH 2 to pH 8. Pressure cooked samples had a protein concentration of 0.5 mg/mL at pH 2 which decreased to around 0.3 mg/mL at pH 4, then increased to 0.65 mg/mL at pH 6 and maintained that level at pH 7 and 8. Boiled samples had a similar shaped curve but with overall lower protein concentrations. Thus, both boiled and pressure-cooked samples had their lowest solubility at pH 4. This agrees with results by [1], who detected an isoelectric point (pI) of 4.6 for centrifuged canned aquafaba and which was similar to the isoelectric point of chickpea protein isolate (pH 4.5). Protein solubility is dependent on the hydrophilic and hydrophobic interactions that balance protein–solvent and protein–protein interactions [23]. Solubility is lowest near the pI because the proteins lack a significant net charge (positive or negative) that provide electrostatic repulsion, thus allowing stronger protein–protein interactions and greater aggregation [24].

### 3.3. Interfacial Properties of Aquafaba

Table 2 shows various interfacial properties of boiled and pressure-cooked aquafaba at ratios of 5:1, 4:1 and 3:1 (water to seed) prepared at pH values of 3, 5 and 7 along with the foaming capacity at the native pH (6). The contribution of the investigated factors and their interactions on these properties is listed in Table 3, which shows the ANOVA output.

#### 3.3.1. Foaming Capacity and Stability

All foaming properties were significantly (*p* < 0.05) affected by pH, which was by far the most influential factor as evident in Table 3. The foaming capacity as well as stability were also significantly (*p* < 0.05) affected by the cooking method. Cooking ratio was not a significant factor for foaming, which aligns with results for protein concentration. Furthermore, while pH did significantly affect foaming volume, only samples at pH 3 had a significantly higher foaming volume than others, except for pressure-cooked aquafaba prepared from a 4:1 ratio. Few significant differences were found among foaming capacities. Pressure-cooked 5:1 and 4:1 aquafaba at pH 3 exhibited the lowest values, significantly (*p* < 0.05) lower than the boiled 3:1 aquafaba at pH 5. As shown in Table 2, all aquafaba samples had significantly lower (*p* ≤ 0.05) protein concentrations at pH 3 which likely led to the decreased foaming capacity. However, samples at pH 3 had significantly greater (*p* ≤ 0.05) foam produced per mg of protein than all other pH values tested. This suggests that, while additional protein did allow for more foam creation, it came with some diminishing returns. A study evaluating the functional properties of 0.1% chickpea protein concentrates showed that samples had highest foaming capacity at pH 2, followed by pH 10 and 8 respectively, with pH values near the pI having the lowest foaming capacity [25]. Other research showed that chickpea aquafaba foaming capacity was not affected by pH, but a different foaming method was used (shaking in an enclosed tube) and it is unclear if samples were centrifuged after the pH was changed [1]. It is possible high variability or overall low foaming capacity could have influenced results.

As Table 2 shows, there were few significant differences in foaming stability after 30 min among the samples. However, a lower pH corresponded to higher foam stability. The pressure-cooked 3:1 sample at pH 3 had significantly higher foaming stability (*p* < 0.05) than several samples at pH 6 or 7 (boiled at ratios 5:1 and 3:1 at pH 6 as well as pressure-cooked 4:1 samples at pH 7 and 5:1 samples at pH 6. The boiled 3:1 and pressure-cooked 4:1 aquafaba at pH 6 and 7, respectively, also had significantly lower foam stability than pressure-cooked samples prepared at ratios of 4:1 or 5:1 and adjusted to either pH 3 or 5. Protein flexibility, concentration and charge are all affected by pH and can significantly affect the foaming properties of proteins. While solubility is diminished, foam stability is improved around the pI but here due to minimized electrostatic repulsion that allows a thick layer of protein to absorb at the air-water interface that helps stabilize the foam [26,27]. Buhl et al. [1] also demonstrated that pH significantly affected the foam stability of canned chickpea aquafaba, with the foams produced at pH 4.5 remaining stable for up to 1 h.

#### 3.3.2. Emulsifying Activity and Stability Indices

The EAI of aquafaba samples ranged from 17.5 m^2^/g to 92 m^2^/g, with the pressure-cooked 5:1 sample at pH 3 producing the highest EAI and the boiled 3:1 sample at pH 7 producing the lowest. Additionally, both pH and cooking method were found to significantly affect the EAI. Analogous to the foaming properties, the influence of pH dominated over other factors or their interactions (Table 3). This appears to be directly related to the protein content of the aquafaba solutions. As displayed in Table 2, samples at pH 3 had significantly low levels of protein, which translated to high EAI values. Overall, this experiment displayed that even though aquafaba only contains low quantities of protein, it can effectively stabilize emulsions. EAI is not easily compared between studies, as differences in method, oil type, oil volume fraction, protein concentration and equipment can all cause significant differences in EAI values and outcome [12]. The EAI of samples at pH 5, 6 and 7 were comparable to those obtained by [1] who had EAI values around 15 m^2^/g which was also significantly higher than that of egg white powder with the same protein concentration.

Of the samples and pH values tested, the boiled 4:1 sample at pH 6 had the highest ESI at 22 min and the boiled 4:1 ratio sample at pH 3 had the lowest ESI at 11.6 min. It was determined that pH significantly affected the ESI time, with samples at pH 6 and 7 having significantly higher values (*p* < 0.05) than those at pH 3 and 5. These results are comparable and consistent with the results obtained by [1] where canned chickpea aquafaba showed an increase of ESI from 15 min up to 25 min with an increase in pH and with pH 7 and 8.5 having a significantly higher (*p* ≤ 0.05) ESI than pH 3 and 4.5. The ESI of multiple legume isolates have been shown to be pH dependent, typically attributed to their higher surface charge and solubility [28]. Additionally, protein composition can impact emulsion stabilization. Due to their full and partial solubility in water, globulins and albumins are more suited for interface stabilization than legume prolamins and glutelins. In a study on rapeseed, their albumins and globulins were found to interact synergistically, with albumins adsorbed to the oil-water interface and globulins weakly bound around the albumins [29]. As was demonstrated by the SDS-PAGE in Figure 1, all the identified protein bands in chickpea aquafaba were albumins or globulins.

### 3.4. Properties of Aquafaba Foams with Added Hydrocolloids

Model systems containing aquafaba, hydrocolloid gums and sugar were also analyzed. These mixtures would be more reminiscent of a meringue, whipped topping or material used in a foam-based cake. This allowed us to test whether these additional ingredients would affect the performance of the foam, assessed by measuring overrun, liquid drainage at two time points, flow properties and texture. As previous results showed that cooking ratio was not a significant factor for foam capacity or stability, only the most concentrated solutions (that is from the 3:1 water to seed ratio) were used. As the chickpea cooking method was a significant factor, aquafaba from both boiling and pressure-cooking was used. As aquafaba at pH 5 was significantly higher (*p* ≤ 0.05) than at pH 3 in foaming capacity and had significantly higher foaming stability than pH 6 and 7 (Table 2), all samples were analyzed at this pH.

#### 3.4.1. Foam Overrun and Liquid Drainage

Table 4 shows the overrun of boiled and pressure-cooked aquafaba with XG and HPMC added at 0%, 0.2%, 0.4% and 0.6% by mass. It ranged from 66.2% to 421%, with pressure-cooked samples with no gum added exhibiting the lowest overrun and boiled samples with 0.4% as well as pressure-cooked samples with 0.6% HPMC having the highest overrun. Cooking method and gum type (*p* ≤ 0.05 for both), but not gum level were shown to significantly affect foam overrun. Interestingly, in pressure-cooked samples the effect of gum addition was more evident. The boiled aquafaba samples all exhibited similar overrun between 351–421%. HPMC addition produced significantly higher (*p* ≤ 0.05) overrun than XG. Boiled and pressure-cooked samples with HPMC did not significantly differ in their overrun values.

The pressure-cooked aquafaba showed a loss of functionality in comparison to boiled aquafaba, with it either unable to properly stabilize the air–water interface or perhaps unable to support the increase in bulk viscosity of the continuous aqueous layer. The low overrun of pressure-cooked aquafaba could, however, be compensated by adding gums. Except for samples with 0.4% XG, there were no significant differences between boiled and pressure-cooked samples when they contained the same gum at the same level. This suggests that the addition of certain stabilizers may be able to counteract variability in raw materials and thereby allow for more consistent outcomes. This could be an advantage in an industrial setting. While XG did significantly improve the overrun of pressure-cooked samples, those samples still had lower overrun than those with added HPMC. XG is not considered to be surface-active [30]. In solutions with limited thermodynamic compatibility, conditions that inhibit proteins and polysaccharides from forming a complex or associating, polysaccharides can cause the protein to perform as a more concentrated film as well as increase surface pressure via exclusion volume effects thereby giving a similar effect to that of increasing protein concentration [30,31]. XG also increases the viscosity of the continuous aqueous phase through water binding. While beneficial to foam stability by reducing drainage, this quality can also inhibit air incorporation and restrict the mobility of surface-active molecules [32]. This could be why boiled aquafaba showed a slight decrease in overrun with an increase in XG levels and why increased levels of XG with pressure-cooked samples did not improve the overrun. On the other hand, HPMC is a surface-active polysaccharide [30]. For surface active polysaccharides, one of two processes may occur. Polysaccharides can adsorb at the air-water interface and be in competition with protein or the surface-active polysaccharide can form a complex with the adsorbed protein through electrostatic interactions or hydrogen bonding [30]. As this experiment was done at pH 5, the charges of HPMC were likely minimal, which could limit its surfactant behavior. Polysaccharides with and without surface activity are able to complex with adsorbed protein, however, and neutral complexes have been shown to form denser viscoelastic interfacial films, therefore contributing to improved foaming properties [30].

As seen in Table 4, egg white had an overrun of 225%. This value was comparable to foams from pressure-cooked aquafaba samples with xanthan gum, but considerably lower than in aquafaba samples containing HPMC, regardless of level.

After 1 h, only three of the samples had any measurable liquid drainage and pressure-cooked samples with no gum displaying the highest value of 18.3 mL. After 24 h, both boiled and pressure-cooked samples without gums had high levels of liquid drainage at 61.3 to 75 mL. The drainage of egg white samples also fell within this range. As with foam overrun, the average drainage of egg white was greater than for pressure-cooked aquafaba samples and lesser than that of boiled samples. Both gum type and addition level significantly affected the amount of drained liquid after 24 h (*p* < 0.05), with xanthan gum being particularly effective at reducing it. For both drainage after 24 h as well as foam overrun the type of gum impacted the results the most (Table 5). There was also a significant interaction effect between gum type and addition level on drainage though it was a comparatively minor factor relative to gum type or addition level individually. The significant interaction was due to all levels of XG essentially eliminating liquid drainage regardless of cooking method. XG’s ability to bind water increases the bulk viscosity of the aqueous phase and this maintains foam stability. On the other hand, HPMC levels did lead to observable differences after 24 h, with the lowest level of HPMC exhibiting significantly higher (*p* < 0.05) amounts of drainage than the other two levels. Liquid drainage is reduced by increasing the viscosity of the solution or the disjoining pressure, i.e., the force per area needed to separate two bubbles separated by the liquid lamella. As HPMC does not significantly increase the viscosity of aquafaba at the experimental temperatures, greater HPMC levels are most likely to have increased the disjoining pressure. The disjoining pressure is associated with the development of osmotic pressure differences between the bulk phase and the lamella fluid and, as such, protruding surfactant chains, counter ion clouds around surfactant layers and hydration repulsion forces all contribute to these pressure differences [33].

#### 3.4.2. Foam Texture

Hardness of aquafaba foams ranged from 0.74 N to 2.02 N and the range of adhesiveness was from 24.3 N·s to 57.2 N·s (Table 4). Gum type and addition level significantly affected (*p* < 0.05) the hardness as well as the adhesiveness, while cooking method was not a significant factor (Table 6). There was a dose-dependent relationship between textural parameters and gum addition level. However, systems with HPMC were affected differently by an increase in gum level than those with xanthan gum. Thus, there was also a significant interaction effect (*p* < 0.05) between gum type and level for both hardness and adhesiveness (Table 6). Foams of both pressure-cooked and boiled aquafaba containing 0.2% HPMC were significantly lower (*p* < 0.5) in hardness and adhesiveness than when 0.4 or 0.6% HPMC had been added. In contrast, hardness and adhesiveness were not significantly different in samples with xanthan gum, irrespective of the level or cooking method. Another crucial difference based on gum type was that at the maximum XG level (0.6%), the hardness of boiled aquafaba foam was no different than that without XG, while foams with 0.6% HPMC were harder than boiled samples without HPMC. Pressure-cooked aquafaba foams without gums produced the softest foams, while both boiled and pressure-cooked aquafaba foams with 0.6% HPMC produced the hardest foams. Pressure-cooked aquafaba with no gums produced foams that were less hard (0.74 N) than those made from boiled aquafaba with no gums (1.47 N). Interestingly, for boiled aquafaba samples, the hardness was lower at 0.2% XG and HPMC than with no added gums, but higher at 0.4 or 0.6% levels. Egg white had a hardness value (1.48 N) similar to that of boiled aquafaba with no added gum as well as boiled aquafaba with 0.6% XG.

The adhesiveness of aquafaba foams exhibited similar trends to those found for hardness, with the boiled aquafaba foam with 0.2% HPMC resulting in the lowest adhesion and both the boiled and pressure-cooked samples with 0.6% HPMC having the greatest adhesion. The addition of 0.2% XG or HPMC had the same effects on boiled aquafaba samples as it did on foam hardness and consistency, i.e., adhesiveness values were significantly lower than that of boiled samples with no gum. As HPMC levels increased, so did the adhesiveness of boiled samples, with 0.6% addition resulting in higher adhesiveness than boiled samples with no gum. Maximum levels of XG in boiled samples resulted in similar adhesiveness to boiled samples with no gum. Lastly, pressure-cooked samples had increasing adhesiveness with higher levels of HPMC and XG, with 0.6% HPMC resulting in a value similar to boiled samples with 0.6% HPMC. All levels of XG in pressure-cooked samples were statistically similar to boiled samples with 0.4 and 0.6% XG. Egg white foam hardness and consistency had values most similar to the boiled aquafaba with 0.6% XG, although they were more adhesive with values most similar to boiled or pressured-cooked samples with 0.6% HPMC.

Regardless of cooking type, 0.6% HPMC resulted in the highest values for hardness and adhesiveness. This is likely due to its complex concentration-dependent behavior, its ability to alter and/or overtake the interfacial characteristics of the adsorbed protein layer and its water binding capabilities [30,34]. The initial decrease in hardness, consistency and adhesiveness observed at 0.2% HPMC and XG could be due to the adsorption of neutral protein-polysaccharide complexes at the interface. These complexes can form dense second layers that prevent proteins from moving through them to form a protein layer that stabilizes the air–water interface [30]. Alternatively, it could be that the proteins and polysaccharides are competitively adsorbing at the interface, making it so the surface pressure is being primarily affected by the component that is able to adsorb more rapidly (i.e., is more surface active) therefore affecting foam structure [30].

#### 3.4.3. Viscosity of Aquafaba/Gum Blends

Table 4 also shows the apparent viscosity and flow behavior index (n) for egg white along with boiled and pressure-cooked aquafaba with sugar and XG and HPMC at levels of 0%, 0.2%, 0.4% and 0.6%. The apparent viscosity ranged from 137 mPa·s to 3339 mPa·s. All samples were much more viscous than samples with no added sugar or gums (Table 4). Gum type and level, as well as their interaction, were significant factors for the viscosity and the flow behavior index (n), but their contribution varied in that gum type was the dominant factor (Table 6). Cooking method was not a significant factor for viscosity or n (Table 6). Overall, XG provided the most pronounced effect on the aquafaba viscosity.

Regardless of cooking method, aquafaba with no gum was not significantly different in viscosity to any sample with 0.2–0.6% HPMC. The samples with the lowest level (0.2% of XG) did not significantly differ in viscosity to each other, to any sample HPMC or to controls without gum. However, boiled as well as pressure-cooked aquafaba with 0.6% XG had a significantly higher viscosity than all other samples, which explains the significant interaction effect between gum type and addition level (Table 6). These results are in line with the literature. HPMC only forms weak thermo-reversible gels upon heating at around 63–80 °C while XG already displays non-Newtonian behavior at room temperature, if present above a critical concentration referred to as the “overlap concentration” [35,36]. As all viscosity tests were performed at 25 °C, samples with HPMC did not show any increased viscosity or changes in pseudoplasticity and were similar to samples with no added gum. Egg white with added sugar had a measured apparent viscosity of 561 mPa·s, which was between the apparent viscosity of boiled aquafaba samples with all levels of HPMC and boiled samples with 0.2% XG.

The fitted value for n ranged from 0.65 to 0.93, with the pressure-cooked aquafaba sample with 0.6% XG having the lowest flow behavior index and therefore greater pseudoplastic (shear-thinning) behavior, while the pressure-cooked sample with no added gum had the highest flow behavior index and thus more Newtonian (linear) behavior. All samples with any level of HPMC had flow behavior index values that were statistically similar (*p* ≥ 0.05), with values all greater than 0.88 indicating more Newtonian behavior. Egg white had a flow behavior index of 0.95, indicating more Newtonian behavior.

## 4. Conclusions

The cooking ratio did not significantly affect the functional properties of aquafaba, while cooking method did affect those properties. This was attributed to the difference in water loss via evaporation leading to a difference in aquafaba concentration. As cooking method and ratio did not influence the protein profile, protein concentration is likely to be what affects functionality regardless of the method of preparation. However, cooking ratio and method were significant factors in the apparent viscosity of aquafaba solutions, implying either an unknown factor such as fiber content or a greater sensitivity to small protein concentration changes. Furthermore, aligning with the results of the protein profile analysis, the cooking method did not affect the protein’s solubility which was lowest around pH 4. While the altering of aquafaba pH significantly affected foaming capacity/stability and the emulsifying activity/stability index, this also was likely primarily influenced by the concentration of protein in the solution. A cooking ratio of 3:1 water to seed of both cooking methods at pH 5 displayed balanced improvement in both foaming capacity and stability.

The evaluated factors (cooking method, gum type, gum level) exerted different effects on foam formation. Gum type, but not its level, significantly affected the overrun. The cooking method also had a significant influence on the overrun, but not on the flow properties or texture. Both gum type and addition level significantly affected the viscosity and the flow behavior index. Depending on the needs of the formed foams, like that of meringue or whipped topping, HPMC and XG can provide a wide range of textural properties, overrun and stability up to 24 h. HPMC-based foams displayed higher levels of overrun overall, significantly improving the low overrun values of pressure-cooked aquafaba. However, HPMC-based foams had greater liquid drainage at 24 h in contrast to XG that had virtually no drainage, although increasing the level of HPMC decreased the overall liquid drainage at 24 h and increased the hardness, consistency and adhesiveness of the foam texture. Essentially, HPMC displays the capability to increase foam texture qualities and the air fraction in chickpea aquafaba with some long-term stability limitations that can be mitigated with higher percent addition. In direct comparison, low levels of XG decreased textural values, with foams with 0.6% XG having similar texture to those with no added gum. The addition of any level of XG can produce highly stable plant-based foams at the expense of its textural qualities. However, different foods or food foams may desire vastly different foam textures. Boiled aquafaba foams, with no gum addition, had higher stability and a greater overrun than that of egg white foams and addition of 0.6% XG led to a similar foam texture. These results indicate that chickpea aquafaba can produce foams with similar likeness to egg white foams and possibly improved qualities of those foams with different levels of either XG or HPMC addition. These results would be most applicable to products such as whipped toppings, meringues, marshmallows or cakes that use an egg-white base foam for volume.

## Figures and Tables

**Figure 1 foods-12-00775-f001:**
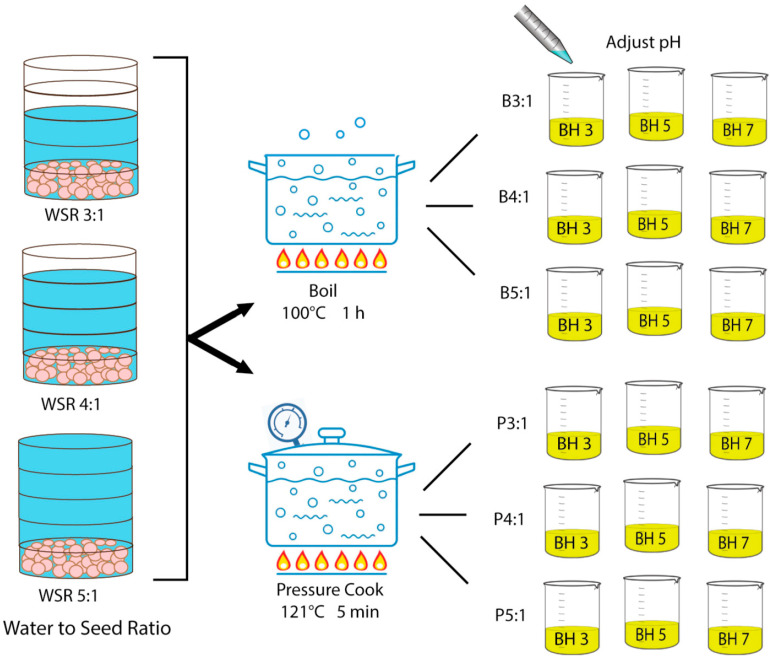
Preparation of chickpea aquafaba at different water-to-seed ratios by boiling or pressure-cooking. Separate samples were adjusted to pH 3, 5 or 7.

**Figure 2 foods-12-00775-f002:**
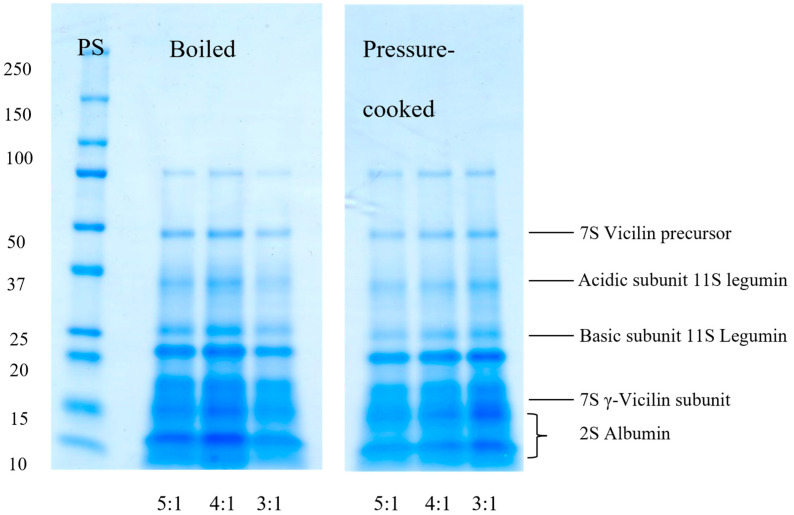
SDS-PAGE gels of aquafaba samples by boiling or pressure-cooking of chickpeas at water:seed cooking ratios of 5:1, 4:1 and 3:1. PS = Protein standard.

**Figure 3 foods-12-00775-f003:**
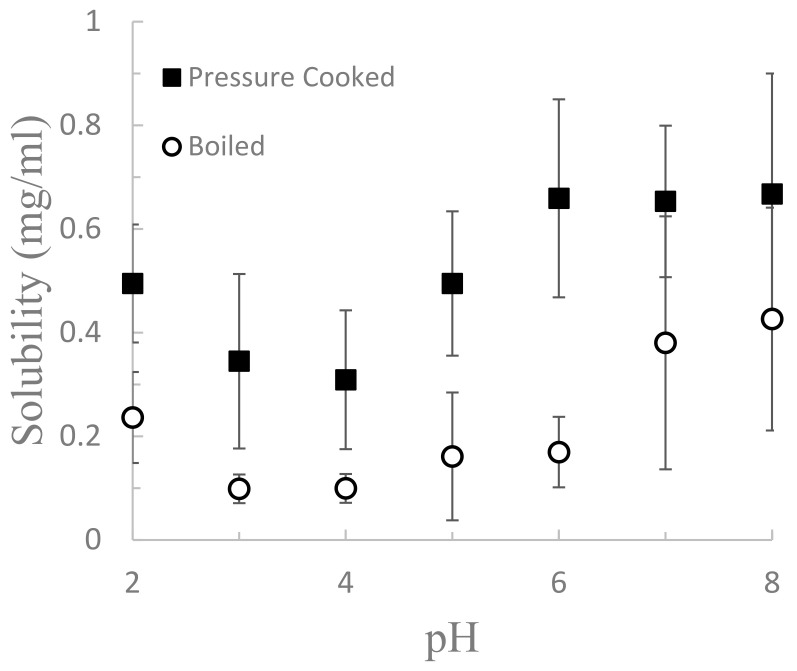
Soluble protein in aquafaba from boiled or pressure-cooked chickpeas adjusted from pH 2 to 8. Error bars represent standard deviations from the mean.

**Table 1 foods-12-00775-t001:** Protein concentration, apparent viscosity (η) and flow behavior index (n) of aquafaba prepared by boiling or pressure cooking at different water to chickpea seed ratios.

Sample ^1^	Protein Concentration (mg/mL)	η (mPa·s)	n
B5:1	1.1 ± 0.1 ^abc^	20 ± 7 ^c^	0.78 ± 0.07 ^b^
B4:1	1.4 ± 0.07 ^ab^	52 ± 4 ^b^	0.70 ± 0.07 ^c^
B3:1	1.3 ± 0.09 ^ab^	81 ± 3 ^a^	0.66 ± 0.03 ^c^
P5:1	0.94 ± 0.1 ^c^	9.6 ± 2 ^c^	0.95 ± 0.4 ^a^
P4:1	0.97 ± 0.1 ^bc^	12 ± 6 ^c^	0.89 ± 0.05 ^a^
P3:1	1.3 ± 0.07 ^ab^	21 ± 9 ^c^	0.81 ± 0.04 ^b^

^1^ Boiled aquafaba indicated by ‘B’. Pressure-cooked aquafaba sample indicated by ‘P’. Water to seed cooking ratio of aquafaba samples indicated by 5:1, 4:1 and 3:1. Mean values (mean value ± standard derivation, n = 9). Values in columns not followed by the same superscript are significantly different (*p* < 0.05).

**Table 2 foods-12-00775-t002:** Protein concentration and interfacial properties of aquafaba samples boiled (‘B’) or pressure-cooked (‘P’) at water to seed ratios of 5:1, 4:1 and 3:1.

Sample	pH	Protein (mg/mL)	Foam Volume (mL/mg)	Foaming Capacity (%)	Foam Stability (%)	Emulsifying Activity (m^2^/g)	Emulsion Stability (min)
B5:1	3	0.47 ± 0.05 ^hi^	34.0 ^bcd^	159 ^bcde^	78.2 ^abcde^	53.9 ^bc^	12.3 ^ef^
B5:1	5	1.1 ± 0.1 ^cdefg^	16.5 ^e^	179 ^abc^	78.7 ^abcde^	31.3 ^cd^	11.9 ^f^
B5:1	6		14.2 ^e^	181 ^ab^	66.3 ^bcde^	21.1 ^d^	21.5 ^a^
B5:1	7	1.5 ± 0.06 ^ab^	11.4 ^e^	169 ^bcde^	68.1 ^abcde^	20.3 ^d^	17.8 ^abcdef^
B4:1	3	0.32 ± 0.1 ^i^	54.0 ^a^	154 ^bcde^	75.8 ^abcde^	71.0 ^ab^	11.6 ^f^
B4:1	5	0.98 ± 0.15 ^defg^	17.4 ^de^	168 ^abcde^	78.9 ^abcde^	28.4 ^cd^	14.5 ^bcdef^
B4:1	6		13.1 ^e^	182 ^abcd^	69.3 ^abcde^	22.5 ^d^	22.0 ^a^
B4:1	7	1.4 ± 0.1 ^abc^	12.8 ^e^	179 ^abcd^	75.3 ^abcde^	19.1 ^d^	16.7 ^abcdef^
B3:1	3	0.35 ± 0.1 ^i^	46.5 ^ab^	158 ^bcde^	80.2 ^abcd^	76.0 ^ab^	13.6 ^cdef^
B3:1	5	1.3 ± 0.1 ^abcde^	15.8 ^e^	200 ^a^	75.8 ^abcde^	23.8 ^d^	14.9 ^abcdef^
B3:1	6		13.9 ^e^	178 ^abcd^	63.7 ^de^	25.3 ^d^	19.2 ^abcd^
B3:1	7	1.6 ± 0.2 ^a^	11.3 ^e^	177 ^abcde^	67.5 ^abcde^	17.5 ^d^	19.8 ^abcd^
P5:1	3	0.26 ± 0.05 ^i^	51.2 ^ab^	133 ^e^	83.5 ^abc^	92.0 ^a^	13.7 ^cdef^
P5:1	5	0.76 ± 0.08 ^gh^	20.5 ^cde^	154 ^bcde^	83.9 ^abc^	29.9 ^cd^	14.1 ^cdef^
P5:1	6		18.3 ^de^	171 ^abcde^	64.6 ^cde^	30.9 ^cd^	18.4 ^abcdef^
P5:1	7	1.2 ± 0.04 ^bcdef^	14.1 ^e^	163 ^abcde^	68.0 ^abcde^	18.5 ^d^	15.3 ^abcdef^
P4:1	3	0.40 ± 0.08 ^hi^	36.0 ^bc^	139 ^de^	85.3 ^ab^	71.7 ^ab^	13.6 ^def^
P4:1	5	0.84 ± 0.1 ^fg^	19.3 ^cde^	162 ^abcde^	84.3 ^ab^	37.1 ^cd^	13.1 ^def^
P4:1	6		19.0 ^cde^	181 ^ab^	69.7 ^abcde^	26.5 ^d^	17.4 ^abcdef^
P4:1	7	1.2 ± 0.2 ^abcde^	14.1 ^e^	172 ^abcde^	59.8 ^e^	22.7 ^d^	20.8 ^abc^
P3:1	3	0.36 ± 0.07 ^i^	40.0 ^ab^	140 ^cde^	86.5 ^a^	79.2 ^ab^	17.8 ^abcdef^
P3:1	5	0.92 ± 0.1 ^efg^	17.7 ^de^	161 ^abcde^	82.0 ^abcd^	32.9 ^cd^	19.5 ^abcde^
P3:1	6		13.5 ^e^	174 ^abcde^	67.9 ^abcde^	22.9 ^d^	18.8 ^abcdef^
P3:1	7	1.3 ± 0.2 ^abcd^	12.2 ^e^	161 ^abcde^	82.2 ^abcd^	20.3 ^d^	17.5 ^abcdef^

Mean values (mean value ± standard derivation, n = 3) for columns with same lower-case superscripts are not significantly different (*p* < 0.05).

**Table 3 foods-12-00775-t003:** Analysis of variance table summarizing the influence of factors (CM, cooking method; WSR, water to seed ratio; and pH) and their interactions on interfacial properties of aquafaba.

	Foam Volume (mL/mg)			Foaming Capacity (%)			Foam Stability (%)			Emulsifying Activity (m^2^/g)			Emulsion Stability (min)		
	Mean Square	*F* Value	Pr > *F*	Mean Square	F Value	Pr > *F*	Mean Square	*F* Value	Pr > *F*	Mean Square	*F* Value	Pr > *F*	Mean Square	*F* Value	Pr > *F*
CM	16	0.419	0.522	3398	20.329	<0.001	278.3	10.541	0.003	811	8.997	0.005	3.63	0.825	0.370
WSR	17	0.426	0.656	145	0.868	0.428	10.2	0.386	0.683	1	0.011	0.989	23.04	5.238	0.010
pH	4406	111.933	<0.001	4660	27.882	<0.001	1212.9	45.945	<0.001	12,993	144.058	<0.001	174.94	39.773	<0.001
CM × WSR	184	4.667	0.016	241	1.444	0.249	8.9	0.337	0.716	203	2.252	0.120	19.14	4.351	0.020
CM × pH	43	1.097	0.345	426	2.549	0.092	45.3	1.716	0.194	136	1.512	0.234	36.37	8.268	0.001
WSR × pH	5	0.124	0.973	170	1.079	0.3814	23.6	0.895	0.477	49	0.540	0.707	12.82	2.914	0.035
CM × WSR × pH	160	4.060	0.008	126	0.756	0.561	5.9	0.225	0.923	281	3.120	0.027	2.09	0.476	0.753
Residuals	39			167			26.4			90			4.40		

Pr—probability.

**Table 4 foods-12-00775-t004:** Foam properties, viscosity (η), shear dependency (n) of aquafaba solutions, along with textural properties (hardness and adhesiveness) of derived foams, containing added xanthan gum (XG) or hydroxypropyl methylcellulose (HPMC) ^2^.

Sample ^1^	pH	Gum	Level (%)	Foam Overrun (%)	Foam Drainage (1 h)	Foam Drainage (24 h)	Hardness (N)	Adhesiveness (N·s)	η (mPa·s)	n
B3:1	5	-	-	409 ^a^	0 ^a^	61 ^b^	1.47 ± 0.07 ^bc^	40.8 ± 3.3 ^bcde^	137 ± 13 ^c^	0.89 ± 0.04 ^a^
B3:1	5	HPMC	0.2	391 ^ab^	1.3 ^b^	66 ^b^	0.94 ± 0.1 ^ef^	24.3 ± 3.2 ^g^	202 ± 9.9 ^c^	0.91 ± 0.006 ^a^
B3:1	5	HPMC	0.4	421 ^a^	3.7 ^c^	47 ^c^	1.76 ± 0.1 ^ab^	47.0 ± 2.9 ^abc^	200 ± 71 ^c^	0.90 ± 0.05 ^a^
B3:1	5	HPMC	0.6	419 ^a^	0 ^a^	37 ^cd^	1.89 ± 0.2 ^a^	56.6 ± 6.4 ^a^	198 ± 12 ^c^	0.91 ± 0.02 ^a^
B3:1	5	XG	0.2	351 ^abcd^	0 ^a^	1.3 ^f^	1.24 ± 0.3 ^cde^	33.4 ± 8.3 ^defg^	734 ± 230 ^bc^	0.72 ± 0.03 ^bc^
B3:1	5	XG	0.4	370 ^abc^	0 ^a^	0 ^f^	1.31 ± 0.2 ^cd^	35.4 ± 7.7 ^cdefg^	1240 ± 420 ^b^	0.68 ± 0.005 ^bc^
B3:1	5	XG	0.6	352 ^abcd^	0 ^a^	0.7 ^f^	1.48 ± 0.08 ^bc^	41.5 ± 3.0 ^bcd^	2820 ± 440 ^a^	0.68 ± 0.07 ^bc^
P3:1	5	-	-	66 ^e^	18 ^d^	75 ^a^	0.74 ± 0.1 ^f^	26.5 ± 7.0 ^fg^	170 ± 52 ^c^	0.91 ± 0.03 ^a^
P3:1	5	HPMC	0.2	375 ^ab^	0 ^a^	64 ^b^	1.11 ± 0.1 ^de^	29.2 ± 2.2 ^efg^	366 ± 43 ^c^	0.93 ± 0.003 ^a^
P3:1	5	HPMC	0.4	401 ^a^	0 ^a^	43 ^c^	1.54 ± 0.3 ^bc^	50.1 ± 12 ^ab^	342 ± 29 ^c^	0.88 ± 0.01 ^a^
P3:1	5	HPMC	0.6	420 ^a^	0 ^a^	18 ^f^	2.02 ± 0.2 ^a^	57.2 ± 6.5 ^a^	207 ± 65 ^c^	0.91 ± 0.02 ^a^
P3:1	5	XG	0.2	279 ^cd^	0 ^a^	0 ^f^	1.28 ± 0.09 ^cd^	35.1 ± 3.3 ^defg^	417 ± 90 ^c^	0.78 ± 0.05 ^b^
P3:1	5	XG	0.4	272 ^d^	0 ^a^	0 ^f^	1.37 ± 0.05 ^cd^	37.5 ± 3.2 ^cdef^	1230 ± 570 ^b^	0.69 ± 0.02 ^bc^
P3:1	5	XG	0.6	302 ^bc^	0 ^a^	0 ^f^	1.35 ± 0.08 ^cd^	38.2 ± 1.5 ^cdef^	3330 ± 330 ^a^	0.65 ± 0.02 ^c^
EW ^3^	-	-	-	225	0	65	1.48 ± 0.1	59.7 ± 2.5	561 ± 390	0.95 ± 0.03

^1^ Boiled aquafaba sample indicated by ‘B’. Pressure cooked aquafaba sample indicated by ‘P’. Aquafaba processed at pH 5 and 3:1 water to seed ratio. ^2^ Mean values (mean value ± standard derivation, n = 3). Values in columns not followed by the same superscript are significantly different (*p* < 0.05). ^3^ Egg white (EW) values included for comparison.

**Table 5 foods-12-00775-t005:** Effect of factors (CM, cooking method; GT, gum type; gum level) and their interactions on aquafaba foam overrun and drainage after 24 h.

	Foam Overrun (%)			Foam Drainage (24 h)		
	Mean Square	*F* Value	Pr > *F*	Mean Square	*F* Value	Pr > *F*
CM	16,272	15.431	<0.001	189	10.825	0.003
GT	62,196	58.982	<0.001	18,655	1068.120	<0.001
Level	1878	1.781	0.190	1043	59.736	<0.001
CM × GT	8663	8.215	0.009	146	8.360	0.008
CM × Level	883	0.837	0.445	58	3.344	0.052
GT × Level	533	0.506	0.609	1023	58.559	<0.001
CM × GT × Level	153	0.145	0.875	58	3.293	0.055
Residuals	1054			17		

Pr—probability.

**Table 6 foods-12-00775-t006:** Relationship between factors (CM, cooking method; GT, gum type; gum level and their interactions) and texture, viscosity (η) and shear dependency (n) of aquafaba foams.

	Hardness (N)			Adhesiveness (N·s)			η (mPa·s)			n		
	Mean Square	*F* Value	Pr > *F*	Mean Square	*F* Value	Pr > *F*	Mean Square	*F* Value	Pr > *F*	Mean Square	*F* Value	Pr > *F*
CM	0.008	0.030	0.864	40.7	1.191	0.279	0.060	0.814	0.376	0.0003	0.307	0.584
GT	0.7456	26.672	<0.001	940.1	27.495	<0.001	17.050	232.281	<0.001	0.3968	383.943	<0.001
Level	1.8174	64.998	<0.001	1990.0	58.203	<0.001	4.705	64.102	<0.001	0.0085	8.200	0.002
CM × GT	0.0055	0.196	0.660	33.1	0.969	0.329	0.005	0.069	0.795	0.0002	0.240	0.629
CM × Level	0.0531	1.898	0.1588	37.2	1.087	0.344	0.085	1.151	0.333	0.0025	2.424	0.110
GT × Level	0.9481	33.907	<0.001	1012.3	29.608	<0.001	5.382	73.322	<0.001	0.0038	3.725	0.039
CM × GT × Level	0.1163	4.160	0.020	3.5	0.102	0.904	0.186	2.535	0.100	0.0010	0.932	0.408
Residuals	0.0280			34.2			0.073			0.001		

Pr—probability.

## Data Availability

Data is contained within the article.

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
