# Peer review of "Evaluation of Processing Conditions and Hydrocolloid Addition on Functional Properties of Aquafaba"

_foods, 2023, doi:10.3390/foods12040775_

Round 1

Reviewer 1 Report

Dear Authors,

 The manuscript (foods-2195132) presented for review concerns evaluation of processing conditions and hydrocolloid addition on functional properties of aquafaba.

Below are the comments I made while reading the manuscript.

1.     Because the paper is similar in subject matter to many already described in the literature. Here are some examples that first appeared in the searched items. The current state of the research field should be reviewed carefully and key publications cited.  Please then describe the novelty elements of the work. The authors cited only one paper (Nguyen et al 2021), while there are more studies with very similar topics.

·       Nguyen, T. M. N., & Tran, G. B. (2021). Evaluation of textural and microstructural properties of vegan aquafaba whipped cream from chickpeas. Chemical Engineering Transactions83, 421-426.

·       Mousavi Kalajahi, S. E. (2021). Investigatingthe effect of different levels of barley aquafaba and xanthan gum as an egg replacer on Physical, chemical, rheological and organoleptic properties of oily cake. Journal of food science and technology (Iran)18(115), 327-338.

·       Nguyen, T. M. N., & Tran, G. B. (2021). Application of Chickpeas Aquafaba with Pre-treatment as Egg Replacer in Cake Production. Chemical Engineering Transactions89, 7-12.

·       I. Arozarena, H. Bertholo, J. Empis, A. Bunger, and I. de Sousa, “Study of the total replacement of egg by white lupine protein, emulsifiers and Xanthan gum in yellow cakes,” Eur. Food Res. Technol., vol. 213, no. 4–5, pp. 312–316, 2001, doi: 10.1007/s002170100391.

·       Solé Lamich, L. (2022). Aquafaba, an egg substitute for food applications.

2.     To better illustrate the combinations used, I encourage you to graphically visualize the process of obtaining samples (section 2.2 and 2.3).

3.     Section 2.3.: Was the pH changed in all samples prepared by the two different heat treatments? It is not explained in this chapter.

4.     Line 154: “A sample (200 µ L) was pipetted from the emulsion into 50 mL of 0.1% w/v sodium dodecyl sulphate (SDS).” Please explain what does sample mean in this section? The description of the methodology is not clear. The methods should be described with sufficient detail to allow others to replicate and build on published results. 

5.     Line 164: What was the assumed value of the V (dilute factor) parameter for the tested systems?

6.     Line 167: how was emulsion dry matter estimated?

7.     Line 168: “mass per unit mass of solvent” - the weight of what?  If proteins, were these values taken from previous analyses? Clarification is needed here.

8.     Line 255: "Total soluble solid"  - Has the determination of this parameter been made?

9.     Line 251: Can viscosity affect the foaming or emulsion process?

10.  Section 2.12. egg white viscosity? Or aquafaba with XG viscosity?

Author Response

Comments and Suggestions for Authors

Dear Authors,

 The manuscript (foods-2195132) presented for review concerns evaluation of processing conditions and hydrocolloid addition on functional properties of aquafaba.

Below are the comments I made while reading the manuscript.

  1. Because the paper is similar in subject matter to many already described in the literature. Here are some examples that first appeared in the searched items. The current state of the research field should be reviewed carefully and key publications cited.  Please then describe the novelty elements of the work. The authors cited only one paper (Nguyen et al 2021), while there are more studies with very similar topics.
  • Nguyen, T. M. N., & Tran, G. B. (2021). Evaluation of textural and microstructural properties of vegan aquafaba whipped cream from chickpeas. Chemical Engineering Transactions83, 421-426.
  • Mousavi Kalajahi, S. E. (2021). Investigating the effect of different levels of barley aquafaba and xanthan gum as an egg replacer on Physical, chemical, rheological and organoleptic properties of oily cake. Journal of food science and technology (Iran)18(115), 327-338.
  • Nguyen, T. M. N., & Tran, G. B. (2021). Application of Chickpeas Aquafaba with Pre-treatment as Egg Replacer in Cake Production. Chemical Engineering Transactions89, 7-12.
  • I. Arozarena, H. Bertholo, J. Empis, A. Bunger, and I. de Sousa, “Study of the total replacement of egg by white lupine protein, emulsifiers and Xanthan gum in yellow cakes,” Eur. Food Res. Technol., vol. 213, no. 4–5, pp. 312–316, 2001, doi: 10.1007/s002170100391.
  • Solé Lamich, L. (2022). Aquafaba, an egg substitute for food applications.

      That is fair point. In our attempt to be concise we omitted details on current work done with aquafaba. We have revised the manuscript (Introduction-yellow highlight) to discuss more of these papers. The primary novelty of our research was that we sought to build on these works by simultaneously considering several elements at once including seed:water ratio, cooking method, pH adjustment and two different hydrocolloids. While this by nature includes some replication of previous studies, we feel that is a good thing. That is, scientific knowledge benefits by assuring that particularly observations are seen in more than one laboratory.

  1. To better illustrate the combinations used, I encourage you to graphically visualize the process of obtaining samples (section 2.2 and 2.3).

As suggested, we have added Figure 1 to the manuscript which visually highlights the steps in preparing water and seeds, cooking and adjusting pH.

  1. Section 2.3.: Was the pH changed in all samples prepared by the two different heat treatments? It is not explained in this chapter.

 Yes, samples prepared both by boiling and pressure cooking had the pH adjusted. This fact has been emphasized to the text.

  1. Line 154: “A sample (200 µ L) was pipetted from the emulsion into 50 mL of 0.1% w/v sodium dodecyl sulphate (SDS).” Please explain what does sample mean in this section? The description of the methodology is not clear. The methods should be described with sufficient detail to allow others to replicate and build on published results. 

      Our apologies if there was any misunderstanding. This passage has been changed to “An aliquot (200 µL) of the freshly prepared emulsion was pipetted into 50 mL of 0.1% w/v sodium dodecyl sulphate (SDS”

  1. Line 164: What was the assumed value of the V (dilute factor) parameter for the tested systems?

The dilution factor is for the general case where the sample is so turbid that it might need to be diluted to bring it into a range for a reasonable absorbance reading. Thus, one might put 1 ml sample and dilute to 10 ml to get a dilution factor of 10. In our case this generally was not required.

  1. Line 167: how was emulsion dry matter estimated? 

As described in He et al. (2019) the emulsion dry matter is determined from the remaining mass after drying the sample overnight at 105°C.

  1. Line 168: “mass per unit mass of solvent” - the weight of what?  If proteins, were these values taken from previous analyses? Clarification is needed here. 

This is perhaps a convoluted way of stating the protein concentration and was carried over from the original papers. A simpler notation is just g protein/g water. This was to create the distinction that it wasn’t a conventional measure such as mg/ml, where one unit is in mass and one in volume, and the volume includes the total volume. This has been changed in the manuscript.

  1. Line 255: "Total soluble solid"  - Has the determination of this parameter been made?

This would be most related to the total protein solubility shown in Figure 2, although we did not dry those samples down to determine the total mass.

  1. Line 251: Can viscosity affect the foaming or emulsion process?

It is fairly well established that viscosity can affect foam and emulsion stability, for example by limiting drainage in the former and dispersed phase movement in the latter. We presume the reviewer is referring to whether the added viscosity can inhibit or enhance the foam or emulsion formation. The answer is likely yes, as added viscosity can reduce the amount of shear and cavitation that helps produce a dispersed phase. We don’t know however to what extent differences in viscosity affected foam or emulsion formation. We had to be satisfied with the empirical observations that foams of a certain volume or emulsions of a particular phase volume were made, and concentrate more on the stability of these.

  1. Section 2.12. egg white viscosity? Or aquafaba with XG viscosity?

Yes, the viscosity of egg white (with sugar) was measured as EW is the traditional standard foaming agent. In addition, as shown in Table 3, the viscosity of aquafaba samples was measured with and without XG or HPMC at concentrations between 0 to 0.6% by mass. We hope this has been clarified in the revised text.

Reviewer 2 Report

This work investigated the effects of processing methods and additives on the functional properties of aquafaba. It is interesting in Food Science and Technology. However, the manuscript is necessary to revise. The detailed comments can be found in the attachment.

Author Response

This work investigated the effects of processing methods and additives on the functional properties of aquafaba. It is interesting in Food Science and Technology. However, the manuscript is necessary to revise. The detailed comments can be found in the attachment.

  1. Pease provide the reason that the samples were processed under such conditions: Is the actual production also carried out under the conditions?

These were chosen as two representative approaches to processing chickpea. At home, or in a commercial setting, one might choose to boil as it is simpler. For canning purposes, one would need to go to high pressure processing/retorting.

  1. There were three cooked samples and three pressure boiled samples, which of them was shown in Figure 2, respectively? Why was the sample chosen?

One sample each from the boiled and pressure-cooked samples were shown in Figure 2. In particular, each of those samples were prepared at. 1:5 seed-to-water ratio. Other seed;water ratios are not shown in the figure as there were no statistical differences due to ratios, just cooking method. The conditions have been highlighted in the revised text (Sec 2.6)

  1. In view of flow behavior index (?), what is innovation of your work when compared with that of Alsalman et al.?

Yes, we were only providing context for what others have found about the rheology of aquafaba, and in particular that aquafaba is typically a shear-thinning fluid, the amount of which depends on the processing conditions. In reality, our n-values were not as low as that for the Asalman (2020) work. There are differences in our approach and objectives. Asalman et al were focused strictly on evaluating the rheological and thermal properties of aquafaba, including measures of both traditional and dynamic (G’,G”) rheology. Their aim was to “optimize” rheological and thermal properties, which we take to mean fit a response surface. While they did examine the influence of water ratio during pressure-cooking, they did not consider boiling. In addition, they did not examine foaming properties. Finally, they worked with conventional-pressure cooking only and did not consider the factors of two different hydrocolloids.

  1. However, I found that the intensities of bands seemed to be significantly different for the different samples according to Figure 1

We were not able to find significant differences in the relative distribution of bands

  1. In view of the profile , what is innovation of your work when compared with that of Alsalman et al. (2021)?

We were not claiming that our approach produced some new and unique combination of proteins. Again, comparison with the work of Ansalman et al (2021) was just to provide context and highlight the types of proteins found in aquafaba. Also, in that study the design was drastically different. They were considering the influence of varying and very high pressure (227-573 MPa) on sophisticated thermal and protein structure. We used a conventional pressure cooker that would reach much lower pressure (0.2 MPa).

  1. Which areas of food may this work have potential applications?

This work would be most applicable to whipped toppings, meringues, cakes with egg white or marshmallows. The manuscript has been revised to include this content.

Reviewer 3 Report

The paper reports the data on  an interesting and well-designed research on the aquafaba properties as influenced by several technological conditions. There are in my opinion few minor points to be improved before the acceptance of the manuscript. In particular:

- As reported in table 1 the protein concentration of aquafaba ranged around 1mg/ml. This concentration is lower than the concentration reported in other studies (1g/100g around). See for example the paper https://ifst.onlinelibrary.wiley.com/doi/epdf/10.1111/ijfs.14427: Could the authors discuss this aspect?

-  Line 242: The quantity of aquafaba obtained through the two processes (BvsP) at a constant water/seed ratio was different if the water losses during the processes is different. The indication of the amount of aquafaba obtained after the trials could be useful to explain the differences observed in Table 1.

- Table 1: Are the authors sure of the standard deviation of B4.1 for the Protein concentration? A SD of 0.7 for a mean value of 1.4 and n=9 is very hight. 

- The statistical significance of the single variables examined should be added in all the tables to better appreciate contribute of process parameters in the variations. For example in Table 1 two lines with the indication of the p-value of the water/seed ratio and of the cooking methods should be added. In table 2 three lines with the indication of the p-value for the water/seed ratio, the cooking methods and the pH should be added, and so on…..

Author Response

The paper reports the data on  an interesting and well-designed research on the aquafaba properties as influenced by several technological conditions. There are in my opinion few minor points to be improved before the acceptance of the manuscript. In particular:

  1. As reported in table 1 the protein concentration of aquafaba ranged around 1mg/ml. This concentration is lower than the concentration reported in other studies (1g/100g around). See for example the paper https://ifst.onlinelibrary.wiley.com/doi/epdf/10.1111/ijfs.14427: Could the authors discuss this aspect?

Yes, the values for our protein do not match all of those in the literature. First, there is the issue of units. Many authors report in g protein/100g sample, some in g protein/100 g dry weight. We choose the units of mg/ml as this was the direct measure from the Bradford assay. Thus, we reported values of 0.94 to 1.4 mg/ml. Roughly speaking, this would be between 0.042 and 0.14 g/100g. As the reviewer notes, many report values of ~1 g/100g. A review by Sole-Lamich (2022) lists values between 0.48 and 1.27 g/100g, so there is quite some disparity. However, and as those authors note, the protein can be measured in several ways, typically by Kjeldahl, Dumas or Bradford assay. Thus, the paper cited by the reviewer (Raikis et al, 2020) uses the Dumas combustion method which can determine total protein. In contrast, we intentionally centrifuged our samples to remove insoluble protein as we felt the soluble protein was most relevant to foaming properties. And as Sole-Lamich indicates, chickpea tends to have less soluble protein than other lentil proteins. Now it is true that Lafarga (2019) shows levels as high as 0.48 mg/100 ml using a Bradford assay. That is the equivalent of 0.0048 mg/ml, and much less than what we found. They also do not mention that the sample was filtered or centrifuged. If not, one might also expect some interference from scattering rather than absorption. In any event, we edited the manuscript to emphasize this is soluble protein and not total protein in the aquafaba.

  1. Line 242: The quantity of aquafaba obtained through the two processes (BvsP) at a constant water/seed ratio was different if the water losses during the processes is different. The indication of the amount of aquafaba obtained after the trials could be useful to explain the differences observed in Table 1.

As stated in the text, the two processing methods did result in different amounts of being lost. It was not large amounts though. Unfortunately, we did not do an exact accounting of those amounts. We might go back and recreate if there is enough time.

  1. Table 1: Are the authors sure of the standard deviation of B4.1 for the Protein concentration? A SD of 0.7 for a mean value of 1.4 and n=9 is very high. 

Thanks for the observation. In fact, the 0.7 had been copied over incorrectly and should have been 0.07. Also, there was a rounding error in one of the other values.

  1. The statistical significance of the single variables examined should be added in all the tables to better appreciate contribute of process parameters in the variations. For example in Table 1 two lines with the indication of the p-value of the water/seed ratio and of the cooking methods should be added. In table 2 three lines with the indication of the p-value for the water/seed ratio, the cooking methods and the pH should be added, and so on…..

We agree this is an important issue. Thus, we have encapsulated the processing and ingredients using an ANOVA Table (Tables 3, 5 and 6). This shows the p- and F- values for each variable and their interaction terms, and how these determine foam volume/stability and emulsion capacity/stability.

Round 2

Reviewer 1 Report

Manuscript has been corrected. I have no further comments.

Reviewer 2 Report

The authors have well revised their manuscript. It can be considered to accept for publication in Foods.